# Control System of Liquid Fertilizer Variable-Rate Fertilization Based on Beetle Antennae Search Algorithm

Jinbin Bai 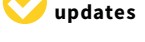, Min Tian * and Jiangquan Li

College of Mechanical and Electrical Engineering, Shihezi University, Shihezi 832000, China;
bajinbin@stu.shzu.edu.cn (J.B.); ljq_mac@shzu.edu.cn (J.L.)
* Correspondence: tm_inf@shzu.edu.cn

**Abstract:** In order to solve the problems of low precision of variable-rate fertilization and uneven fertilization flow of field liquid fertilizer applicator, a control system of variable-rate fertilization of liquid fertilizer based on beetle antennae search algorithm was proposed. First of all, this study established a mathematical model for the variable-rate fertilization control system of liquid fertilizer. Then, according to the control requirements, the search algorithm is used to optimize the three parameters of Proportion Integration Differentiation (PID). Finally, the response time and overshoot of the system are analyzed by software simulation, and the PID control based on beetle antennae search algorithm is compared and analyzed with fuzzy PID control and traditional PID control. The control effect of the control system is verified by a bench test. The results show that the actual response time of the variable-rate fertilization control system based on the beetle antennae search algorithm can reach 2 s, and the average relative error can reach 1.27%. Therefore, the control system of this study can achieve a better control effect, and the control method of this study provides a feasible scheme for the study of variable-rate fertilization.

**Keywords:** beetle antennae search algorithm; control system; variable-rate fertilization; liquid fertilizer; flow

## 1. Introduction

Scientific and reasonable fertilization plays an extremely important role in improving crop yield, saving water resources, and improving soil quality. Therefore, it is very necessary to achieve the sustainable development of precision agriculture. Precision agriculture is developed from the field of agricultural production and an interdisciplinary field of study at home and abroad has been widely recognized [1–4].

The purpose of precision agriculture is to improve the yield, quality, and income of crops by optimizing the input of water and fertilizer in agriculture [5,6]. Avoiding waste water and fertilizer is one of the important purposes of precision agriculture. Variable-rate fertilization, as a key technology and the basis of precision agriculture, provides a new and effective way to meet the needs of modern agriculture's "green planting", input on-demand, and uniform fertilization. Variable-rate fertilization technology can improve fertilizer utilization and reduce the impact on the environment [7,8]. Variable-rate fertilization aims to apply specific and accurate fertilizers at different locations to meet the requirements of water and fertilizer management at specific locations [9,10].

In recent years, precision agriculture and variable-rate fertilization are developing rapidly. The soil area division methods are gradually changing by sensor technology [11,12], and the variable-rate fertilization method is also gradually developing towards automation and intelligence [13].

Ning Su et al. [14]. proposed a new variable-speed fertilization system based on adjusting the length of the active feed roller of the grooved roller. The fertilization system is designed for mechanical structure and actuator, and is designed to be a low-cost, stable

embedded decision system and corresponding software. The system uses the SpatiaLite database to solve the problem of spatial location search and spatial data query, and verifies the dynamic adjustment and stability of the system. However, the response time of the system in this study is longer, and for the fertilization process of solid fertilizer, the accuracy of fertilization is not improved much.

Yinyan Shi et al. [15]. developed a device to adjust the amount of fertilizer by using a driver to improve the rapid response ability of the variable-speed fertilization control system of the centrifugal variable-speed seeder and reduce the error corresponding to the feed fertilization system. This study analyzed the response delay of the delayed fertilization model of the system, and then tested its performance. The experimental results show that the variable-speed fertilization control system with modified response time alleviates the problem of fertilization lag and improves the response speed of rice variable-speed fertilization. However, the average error of the system is more than 9%, which is not a good result for the liquid fertilizer system.

In order to realize the precise control of the bivariate control system of the variable-rate rate applicator, Jiqin Zhang et al. [16]. proposed an optimization method that combines a differential evolution (DE) algorithm with a multi-objective evolution algorithm (MOEA/D) of decomposition algorithms. The results show that the optimization method is superior to the traditional optimization method, which can reduce the average relative error by about 25% and limit the response time to about 2 s. However, in practical operation, the system needs to calculate the optimal control sequence that takes a long time online, which will affect the real-time performance of the fertilization system.

Da Silva M J et al. [17]. developed a site-limited nitrogen management injection dosing system. Variable-rate fertilization with synchronous liquid fertilizer injection and soil drilling is the central idea of the application in design. The study can effectively reduce the loss of fertilization, but lacks verification of the actual effect of the device.

Mendes W R et al. [18]. developed an intelligent fuzzy reasoning system based on precise irrigation knowledge, which can create a standard map to control the rotation speed of the central pivot. The results show that the uncertainty and nonlinear problems of irrigation systems can be solved when the data from environmental variables are well fitted with fuzzy logic, and the control model of high precision irrigation is established. However, this study did not conduct field experiments and failed to effectively evaluate the actual performance of the system.

Reyes J F et al. [19]. conducted a field test on an automatic control system for variable-rate fertilization. This study verified the error and dynamic response effect of the fertilization system under different fertilization widths. However, the systems involved in this study are mainly functional, and there is a lack of innovative research in the article.

Zhang J. et al. [20]. designed a variable-rate liquid fertilizer control system to meet the requirements of precise fertilization. The control system can realize the real-time ratio and mixing of N, P, and K liquid fertilizers according to the decision support subsystem. The system adopts segmented integral separation PID to realize the precise control of liquid fertilizer. In addition, it was verified by experiments that the error of fertilization and fertilization in the system was 5% and the response time was 6 s. However, the response speed of the system is not good enough, and the response of the system can be optimized by fertilization compensation and other methods.

Qi J. et al. [21]. designed a subsoiling variable-rate fertilization machine based on conservation tillage and precision agriculture. This equipment can complete multiple working procedures (including subsoiling and variable-rate fertilization) in one operation. At the same time, the field experiments verified that the subsoiling performance and variable-speed fertilization index of the machine met the requirements of GB/T 24675.2-2009. However, this study failed to achieve a comprehensive application of subsoiling and variable-rate fertilization.

Xiuyun X et al. [22]. proposed a variable-rate liquid fertilizer applicator for deep fertilization based on ZigBee technology. The experiment verifies that the precision of

the fertilizer applicator can reach 99.52%. However, the experiment did not consider different fertilization environments, and there was no in-depth study on variable control of systematic pressure. Jiang Y. et al. [23]. improved the structure of the Venturi variable-rate fertilization device based on pulse width modulation, and a closed-loop mixed fertilizer control system based on fuzzy control algorithm is designed. However, the study lacks an experimental platform and does not consider the impact of the actual environment.

According to the above research, there are many research directions for variable-rate fertilization. Both the improvement of mechanical structure and the optimization of the control process are aimed at improving the accuracy of fertilization and improving the utilization rate of fertilizer. Among them, many scholars use the PID algorithm, PWM control, or fuzzy control to study the control system. In the self-propelled or traction spray applicator [24], the speed of flow detection feedback in the pipeline and the response time of the electric proportional valve to adjust the valve opening according to the demand are important factors to be considered in the variable-rate fertilization control system [25,26].

In order to improve the fertilization accuracy of liquid fertilizer and improve the response speed of the control system, this paper proposes a variable-rate fertilization control system based on beetle antennae search algorithm. The system adopts the beetle antennae search algorithm to adaptively optimize the parameters of the PID algorithm, simulation, and bench test.

## 2. Structure and Working Principle of Fertilizer Applicator

The overall structure of the high-altitude traction liquid fertilizer applicator in this paper is composed of a high-altitude traction frame, self-priming spray pump, liquid fertilizer spray rod, liquid fertilizer box, spray rod suspension, spray rod fixing frame, control valve group, and fertilization pipeline, as shown in Figure 1. The liquid fertilizer variable-rate fertilization control system is mounted on a high-clearance tractor. The operating width of the liquid fertilizer spray rod is 10 m, and the tractor speed varies from 1 to 3 m/s. Fertilizer spray bar consists of four sections, the middle of the liquid fertilizer spray bar is spray bar suspension, the suspension is installed on the spray bar fixed frame, the suspension is supported by the shock absorber and hydraulic rod, the suspension of the fertilizer height can be adjusted in 0.7~0.92 m. The fertilizer spraying rod and spraying rod, and the fertilizer suspension and fertilizer spraying rod are connected by hinges, which can be folded or extended in the horizontal direction. The spray nozzles are installed below the fertilizer spray rod and evenly distributed in the horizontal direction. Three sprinklers are fixed at the lower end of each section of the fertilization boom in the vertical direction, and four sprinklers are fixed under the fertilization suspension. The control valve group and the self-priming jet pump are installed in the front section of the traction frame. The solenoid valve is used to control whether or not to spray fertilizer under each section of the fertilizer spray rod. Electric proportional valves are used to control the flow of the mains. This fertilizer applicator performs spray fertilization with the pressure provided by a self-priming jet pump.

The device adopts spray rod spray water and fertilizer integrated fertilization method. The important part of the device to control flow is the electric proportional valve, which is installed in the valve group in Figure 1. The segmented valve in the valve group consists of five solenoid valves, which only control the opening and closing of the valve. According to the demand of liquid fertilizer variable-rate fertilization, the principle of the system control flow is to obtain the fertilizer amount required for the current field according to the fertilization prescription diagram, and then the opening of the electric proportional valve changes with the change of vehicle speed. The fertilizer amount is detected by the flowmeter and fed back to the controller. The controller analyzes and compares the real-time flow, the fertilizer amount required for the current field and the real-time vehicle speed, thus forming a closed-loop negative feedback regulation system to ensure the uniformity of fertilization in the current field. The stability and system response accuracy of the

closed-loop negative feedback control are the key factors affecting the fertilization accuracy of the liquid fertilizer variable-rate fertilization control system in this paper.

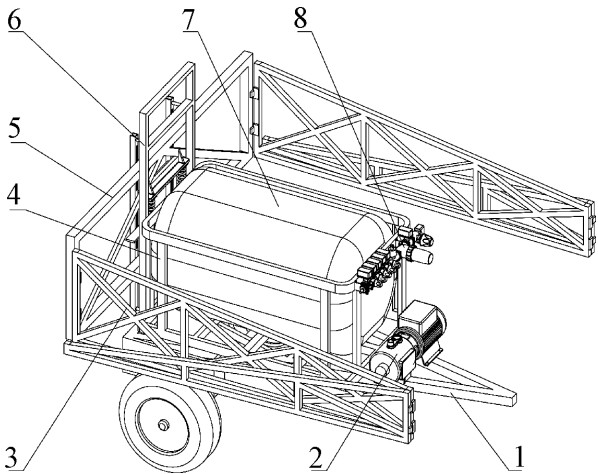

**Figure 1.** Overall structure of traction liquid fertilizer sprayer. 1—Traction frame. 2—Self priming jet pump. 3—Fertilizer spray rod. 4—Fertilizer tank bracket. 5—Spray rod suspension. 6—Spray rod fixing frame. 7—Liquid fertilizer box. 8—Valve group.

The main structure and performance parameters of the liquid fertilizer applicator are shown in Table 1.

**Table 1.** Main structure and performance parameters of liquid fertilizer sprayer.

| Parameters | Values |
|---|---|
| Overall dimensions (Folded state)/(m × m × m) | 2.16 × 2.20 × 1.78 |
| Height of frame base from ground/m | 0.526 |
| Fertilizer tank/(m × m × m) | 1.12 × 0.60 × 0.60 |
| Working width/m | 10 |
| Working height/m | 0.7~0.92 |
| Vehicle speed/(m·s$^{-1}$) | 1~3 |
| Head range of pump/m | 5~50 |

## 3. Liquid Fertilizer Variable-Rate Fertilization Control System Modeling

In order to optimize the control process of the liquid fertilizer fertilization system and reduce the response time of the control system, the transfer function model of the liquid fertilizer fertilization control system is established. The control model of the liquid fertilizer variable-rate fertilization control system in this paper is based on the real-time vehicle speed collected by the angular velocity sensor as the input. After conversion, the controller sends the electrical signal to the electric proportional valve, which controls the valve opening. Finally, the system output is the flow of liquid fertilizer. The flow is fed back to the controller through the flow meter of the control system block diagram in Figure 2, and the closed-loop negative feedback control is carried out by the controller.

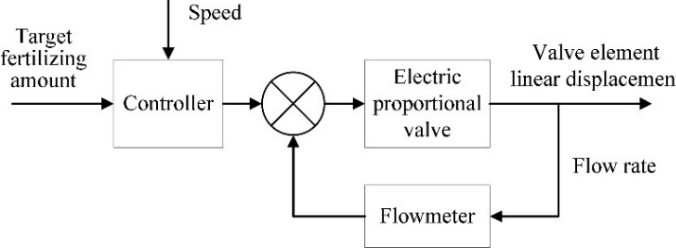

**Figure 2.** Block diagram of liquid fertilizer variable-rate rate fertilization control system.

According to the input and output relationship of the control system block diagram, the input and output relationship of the system can be obtained as follows:

$$Q = f(v, \lambda) = \lambda \cdot 10^{-4} \cdot l \cdot v, \tag{1}$$

$Q$ is the volume of liquid fertilizer output by the variable-rate fertilization control system, L/min. $Q$ is a process function. $v$ for vehicle speed, m/s. $\lambda$ is the target fertilizer rate of the input system. According to the amount of foliar fertilizer in the six cotton fields of Xinjiang, China, 149 regiment, $\lambda$ is set to 250 L/ha. $l$ represents fertilization width, m. According to this design, $l = 10$ m.

According to Figure 2, the input of the feedback channel of the system is the real-time flow read by the flowmeter, and the signal output from the feedback channel to the controller is the voltage signal. After the signal is converted by the controller, the speed of the input system and the target fertilizer amount are compared and adjusted to achieve the negative feedback control of the control system.

Therefore, the feedback function in the control model can be expressed as follows:

$$H(s) = \frac{v(s)}{Q(s)} = \frac{500}{3 \cdot \lambda \cdot l} \cdot e^{-\tau s}, \tag{2}$$

In the formula, $\tau$ is the signal transmission delay time of the feedback link, s. $s$ is the complex variable after the Laplace transform of the transfer function. $H$ is the negative feedback link of the transfer function.

In this paper, the flow meter detection of the control system is real-time. According to the hardware conditions, the delay time of the feedback link can be ignored.

According to the control requirements of the liquid fertilizer variable-rate fertilization control system, the electric proportional valve is the main control object, and the 463,020 electric proportional valves of the ARAG company are selected. The signal control block diagram is shown in Figure 3.

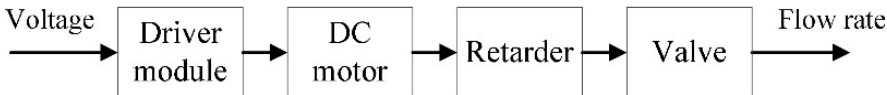

**Figure 3.** Block diagram of 463,020 electric proportional valve control system.

According to Figure 3, the input and output of the drive module are voltage signals, and the transfer function is a proportional link plus a delay link. The relationship is as follows:

$$G_1(s) = \frac{U_{\text{out}}}{U_{\text{in}}} = K_s \cdot e^{\tau s}, \tag{3}$$

where $Ks$ is the amplification factor of the converter. $U_{\text{out}}$ and $U_{\text{in}}$ represent the output and input voltage signals of the drive module, V. $G_1(s)$ is the transfer function of the drive module.

The delay $\tau$ of the voltage signal transmission process of the drive module is less than 0.05 s, and the influence of $\tau$ on the system can be ignored. Therefore, the transfer function of the drive module can be expressed as a proportional link.

In the DC motor of the electric proportional valve, the control signal input is the voltage signal, and the output is the motor shaft angle. The signal control circuit of the DC motor includes the balance of armature circuit, electromagnetic induction of motor rotor, and torque balance of motor shaft. The balanced equation is shown in Formula (4).

$$\begin{cases} u_a(t) - E = i(t) \cdot R_a + L_a \cdot \frac{di(t)}{dt} \\ E = k_\varepsilon \cdot \omega_m = k_\varepsilon \cdot \dot{\theta}_m \\ k_m \cdot i(t) - M_1 = J_m \cdot \dot{\omega}_m = J_m \cdot \ddot{\theta}_m \end{cases}, \tag{4}$$

where $t$ is time, s. $u_a(t)$ is the input voltage of DC motor, V. $e$ is the motor electromotive force, V. $i(t)$ is the armature current, A. $R_a$ is the total armature resistance, $\Omega$. $L_a$ is the total inductance of the armature, H. $k_\varepsilon$ is the back EMF coefficient. $\theta_m$ is the motor shaft angle, ($^\circ$). $k_m$ is the motor torque coefficient. $M_1$ is the motor load torque, N·m, and, where $f$ is the friction coefficient. $J_m$ is the rotational inertia of the motor rotor, kg/m$^2$. $\omega_m$ is the angular velocity of the motor rotor, rad/s.

Laplace transform is performed on Equation (4) to obtain the transfer function of the DC motor of the electric proportional valve, as shown in Equation (5).

$$G_2(s) = \frac{\theta_m(s)}{U_a(s)} = \frac{k_m}{L_a(f + J_m)s^3 + R_a(f + J_m)s^2 + k_m k_\varepsilon s}, \tag{5}$$

where $U_a(s)$ is the Laplace transform function of the input voltage of the motor. $G_2(s)$ is the transfer function of the DC motor.

The reducer is composed of gear sets, and the speed of the DC motor output shaft is output as the displacement of the valve core after passing through the reducer. The displacement X of the spool is 0~19 mm. In this paper, the linear displacement of the spool is the opening of the electric proportional valve.

The input–output signal transmission relationship of reducer is mainly reflected by the transmission ratio. The control process is proportional control, and the transfer function is expressed as follows:

$$G_3(s) = \frac{X(s)}{\theta_m(s)} = \frac{L}{2\pi \cdot i}, \tag{6}$$

In the formula, $i$ is the transmission ratio of the reducer, $L$ is the transmission rod guide, mm; $X(s)$ is the Laplace transform function of spool displacement; $G_3(s)$ is the transfer function of the reducer.

The opening and flow of the electric proportional valve used in this study are linear under the condition of fixed pressure, so the relationship between flow and opening can be expressed by Equation (7).

$$G_4(s) = \frac{Q(s)}{X(s)} = K_q, \tag{7}$$

$G_4(s)$ is the transfer function of valve opening and flow.

According to the control system diagram in Figure 3, the control object of the forward channel is mainly the electric proportional valve, and its transfer function is expressed as

$$G(s) = \frac{Q(s)}{U(s)} = G_1(s) \cdot G_2(s) \cdot G_3(s) \cdot G_4(s), \tag{8}$$

In the formula, $G(s)$ is the transfer function of the electric proportional valve.

In the transfer function, the relevant parameters of the control system are shown in Table 2.

**Table 2.** Parameters of the control system.

| Parameter Name | Value |
|---|---|
| Amplification factor of the converter $k_s$ | 2 |
| Total armature resistance of the electric proportional valve $R_a$ | 2.68 $\Omega$ |
| Total armature inductance of the electric proportional valve $L_a$ | 3.87 mH |
| Motor torque coefficient of the electric proportional valve $k_m$ | 0.0429 N·m/A |
| Back EMF coefficient of the electric proportional valve $k_\varepsilon$ | 0.042 V/(rad/s) |
| Rotational inertia of motor rotor of the electric proportional valve $J_m$ | $1.67 \times 10^{-5}$ kg·m$^2$ |
| Transmission rod guide $L$ | 0.005 m |
| Friction coefficient $f$ | $3.1 \times 10^{-6}$ |
| Transmission ratio of the reducer $i$ | 100 |
| Proportional coefficient of flow and opening $K_q$ | 45 |

According to the control model and the parameters of the control system, the closed-loop feedback control transfer function of the variable-rate fertilization control system is expressed as

$$G_r(s) = \frac{G(s)}{1 + G(s)H(s)} = \frac{1.93}{4.8s^3 + 3.3s^2 + 113s + 28.95}, \tag{9}$$

where $G_r(s)$ is the transfer function of the liquid fertilizer control system.

PID control is widely used in industrial process control due to its simple algorithm, good robustness and high reliability, especially for deterministic control systems that can establish accurate mathematical models. PID controller is based on the error of the system, using the proportion, integral, differential calculation of the control quantity to control.

The formula of PID parameter tuning by PID control algorithm and fuzzy control is as follows:

$$u(k) = K_P \cdot e(k) + K_I \cdot \int_{i=0}^{k} e(i)di + KD \cdot \frac{de(k)}{dk}, \tag{10}$$

in the form $u(k)$ is the output of the control system. $e(k)$ is flow error, L/min. $\int_{i=0}^{k} e(i)di$ is the cumulative error of flow. $de(k)/dk$ is the change rate of flow error; $K_P$, $K_I$, and $K_D$ are the coefficients of proportional, integral, and differential terms, respectively.

## 4. Simulation Analysis of Control System Model

### 4.1. Beetle Antennae Search Algorithm

Beetle antennae search algorithm is a biologically inspired intelligent optimization algorithm, which is inspired by the foraging principle of longicorn.

Beetle antennae search algorithm principle is as follows. When the beetle is foraging for food, the beetle does not know where the food is but forages according to the strength of the food's smell. Long beetles have two long antennae. If the odor received by the left antennae is greater than the right, then the beetle will fly to the left in the next step, otherwise it will fly to the right. According to this principle, the beetle can find food.

Among them, the smell of food is equivalent to a function, which is different at each point in the three-dimensional space. The two beetles can collect the odor value of two points near themselves. The purpose of the beetle is to find the point with the largest global odor value (that is, the location of the food). The intelligent optimization algorithm can be used to solve the function optimization by imitating the behavior of the longhorn beetle.

### 4.2. Beetle Antennae Search Algorithm to Optimize PID Parameters

Appropriate proportional coefficient, integral coefficient, and differential coefficient can effectively reduce the error of the system and reduce the response time of the control system. In this study, the three parameters of the PID controller are optimized by the Taurus search algorithm, and the optimization steps are as follows.

(1) Establish a three-dimensional space problem, each one-dimension array represents proportional coefficient, integral coefficient, and differential coefficient, respectively. The upper limit is set to (100,40,0), and the lower limit is set to (50,15,−1).

(2) Establish an initial spatial matrix based on the above spatial problems and the population size. The population size is set to 15.

(3) The fitness function of the optimization problem is established based on the time multiplied absolute error integral criterion (ITAE), which is expressed as follows.

$$J(\text{ITAE}) = \int_{0}^{\infty} t|e(t)|dt, \tag{11}$$

(4) The direction of optimization is determined by the set step length and the distance between the two strands of the beetle. Among them, the step length is set to 2, the distance between the two whiskers is 2.

(5) Find the PID parameters which can minimize the fitness in the iterative space matrix, and use them as the optimal solution of the problem. Finally, this parameter is

substituted into the control model for simulation analysis to obtain the optimized control waveform of the control system.

The process of optimizing PID parameters by beetle antennae search algorithm is shown in Figure 4.

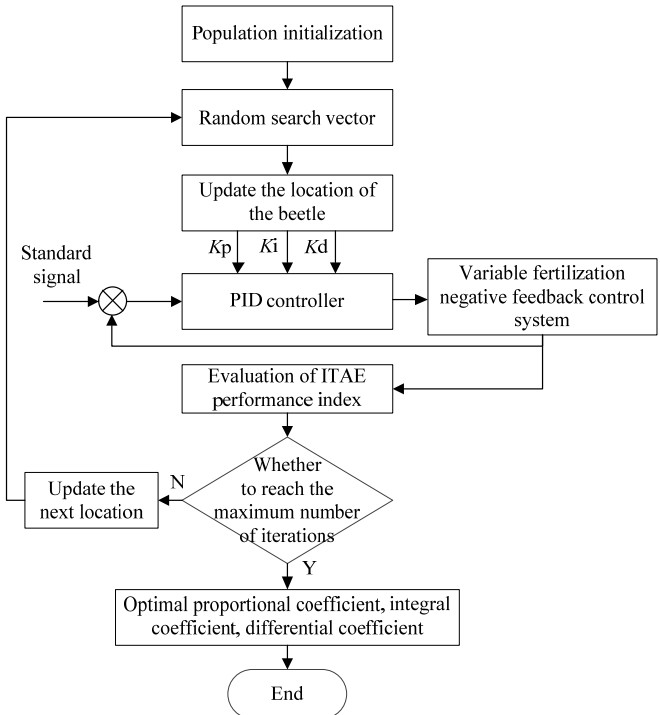

**Figure 4.** Flowchart of optimizing PID parameters by beetle antennae search algorithm.

The location information of the beetle in this paper is represented by a three-dimensional (3D) coordinate. The three coordinates represent the three parameters of the PID controller. The randomly searched vector represents the direction and length from the current 3D coordinate to the next 3D coordinate. When a new 3D coordinate is generated, the algorithm will judge the pros and cons of the PID parameter corresponding to this coordinate and the PID parameter corresponding to the previous coordinate. Then, select a better parameter as a reference for selecting the next 3D coordinate. Finally, the algorithm obtains the optimal value of PID parameters by repeating the above work continuously.

### 4.3. Simulation Analysis of Control System

#### 4.3.1. Simulation of PID Control System

According to the model of the variable-rate fertilization control system in this paper, the simulation software establishes the PID control model of the variable-rate fertilization control system. The amplitude of the input step signal is set to 1, and the parameters of the PID controller are adjusted to analyze the waveform of the system output.

The simulation process of PID control model is as follows: a step signal with an amplitude of 1 is input at $t = 0$, the simulation time is set to 30 s, and the $K_P$, $K_I$, and $K_D$ of PID controller are adjusted, and then the output waveform is sent to the oscilloscope. The simulation waveform is shown in Figure 5.

According to Figure 5, the delay time of the transient response of the system (the time required when the response curve reaches half of the steady-state value for the first time) is 0.73 s. When the error of the control system is stable within 3%, the response time is 2.5 s, and the overshoot is 0.014. According to the empirical trial and error method, $K$p = 72.8, $K$i = 17.9, $K$d = −0.67 are finally selected.

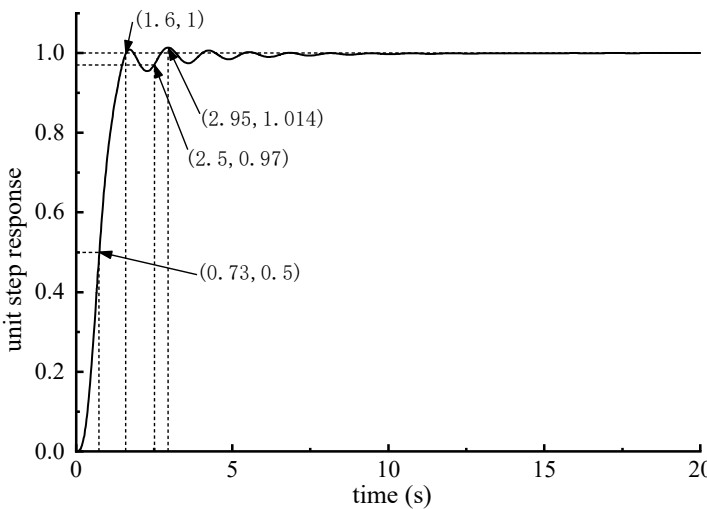

**Figure 5.** Simulation waveform of PID control.

### 4.3.2. Simulation of Fuzzy PID Control System

Fuzzy PID control is based on the PID algorithm, using the error e and the error rate of change $e_c$ as input, using fuzzy rules to carry out fuzzy reasoning, querying the fuzzy matrix table to adjust parameters to meet the self-tuning of PID parameters by $e$ and $e_c$ at different times. Theoretically, fuzzy PID control can realize the self-adaptive adjustment of PID parameters to a certain extent.

The simulation results of fuzzy PID control in the control system of this paper are shown in Figure 6.

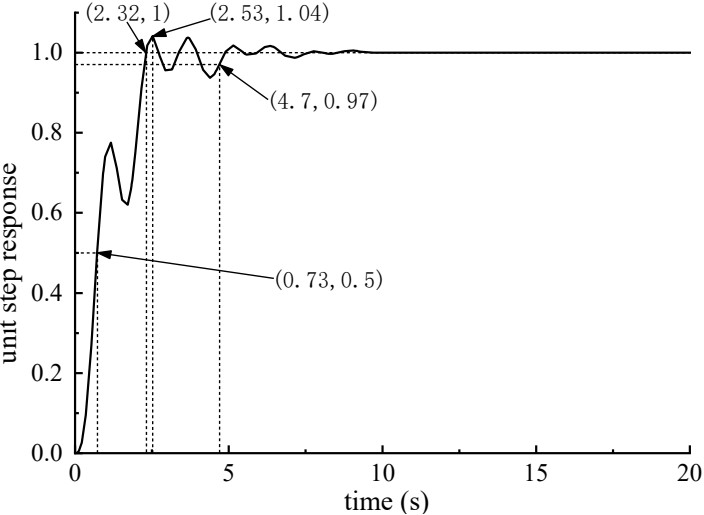

**Figure 6.** Simulation waveform of fuzzy PID control.

According to Figure 6, the response time of fuzzy PID control is 4.7 s when the system error is stable within 3%, which is 2.2 s longer than that of traditional PID control. The overshoot of fuzzy PID control is 0.04, which is 0.26 more than that of traditional PID control. Therefore, it can be seen that the fuzzy PID control can adjust the PID parameters adaptively, but its system response performance is worse than that of the traditional PID control.

### 4.3.3. Simulation of PID Control System Based on Beetle Antennae Search Algorithm

According to the design process of optimizing PID parameters by beetle antennae search algorithm, the simulation model of the control system is established, and the characteristics of the system are judged by the ITAE performance index. The iterative process

of the optimal individual fitness of the control system and the response curve is shown in Figure 7.

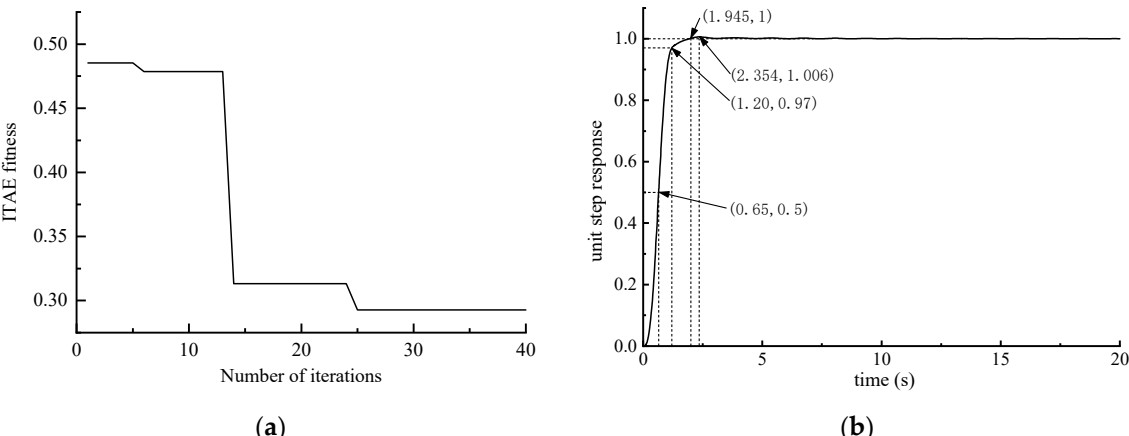

|     |     |
| :-: | :-: |
| (**a**) | (**b**) |

**Figure 7.** The optimized simulation results are as: (**a**) Iterative process of optimal individual fitness; (**b**) Simulation waveform of PID control based on beetle antennae search algorithm.

According to Figure 7b, the delay time of the transient response of the system is 0.65 s. When the error of the control system is stable within 3%, the response time is 1.2 s, and the overshoot is 0.006.

Compared with PID control, the response time of the system after the optimization of the beetle antennae search algorithm is reduced by 1.3 s, and the overshoot is reduced by 0.008. Compared with fuzzy PID, the response time of the system optimized by the beetle antennae search algorithm is reduced by 3 s, and the overshoot is reduced by 0.034. It can be seen that PID control based on beetle antennae search algorithm can make the system achieve faster response and improve system stability.

## 5. Bench Test

### 5.1. Experimental Materials and Platform

Fertilization experiments were conducted in the glass greenhouse of Shihezi University. The test platform includes control valve group ARAG 473, nozzle ARAG 422, filter ARAG 3269113, pipeline, ARAG WOLF impeller flowmeter, electric proportional valve ARAG 463, self-priming jet pump JET 5-50-1.8, controller APC-3072, switch box, etc., as shown in Figure 8. The maximum head of the self-priming ejector is 50 m, the maximum suction is 9 m, and the maximum flow rate is 83.3 L/min. The controller uses Intel AtomTM processor E3845, 4G DDR3L memory, five-wire resistive touch screen. The maximum diameter of the electric proportional valve spool is 19 mm, and the maximum flow rate is 1016 L/min under 0.15 MPa pressure.

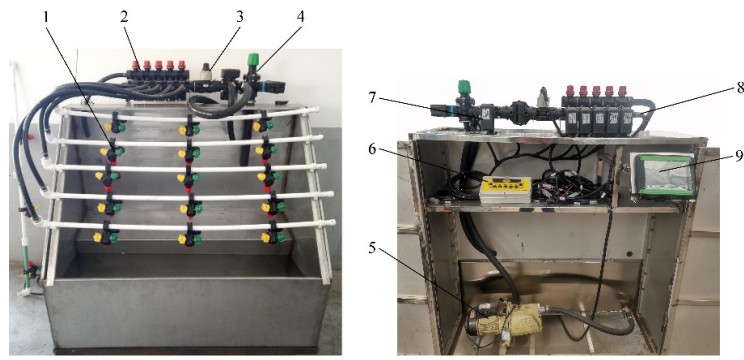

**Figure 8.** Bench test platform. 1—Spray nozzle. 2—Segmented valve group. 3—Flowmeter. 4—Electric main valve. 5—Self-priming jet pump. 6—Switch box. 7—Electric proportional valve. 8—Pressure sensor. 9—Controller.

The control objects in the experiment are the electric proportional valve and solenoid valve, and the test material is clear water without solid suspended particles. The accuracy of fertilization flow control is measured and verified by traditional PID control and PID control based on beetle antennae search algorithm.

The self-priming jet pump of the experimental platform is powered by 220 V AC. The controller and actuator convert 220 V AC to 12 V DC through a switching power supply and then send it to the control system. The test platform is 1.4 m high and 0.6 m wide.

### 5.2. Results and Analysis

### 5.2.1. Stability Experiment of Control System

The data measured by the electromagnetic flowmeter is displayed on the controller screen, and the actual spraying liquid fertilizer flow is measured by experiments. According to the actual driving speed of the field tractor, the speed in the experiment was set to 1.5, 2.0, 2.5, and 3.0 m/s through the controller, and the flow corresponding to each group of speed was measured. The average value of each group of flow was measured five times. Under the condition of fertilization amount set to 250 L/hm$^2$, according to Formula (1), the theoretical fertilization flow under these four different speeds was 22.5, 30.0, 37.5, and 45.0 L/min, respectively.

The flow output of the system under four working conditions is measured. Each working condition is measured three times, and the single measurement duration is 1 min. The average value of three measurements is used as the measurement flow of the working condition. At the same time, the absolute error and relative error of each control algorithm are recorded. The experimental results are shown in Table 2.

Table 3 shows that the average relative error of the traditional PID flow is 4.39%, and the maximum absolute error is 2.06 L/min. The average relative error of PID control based on the beetle antennae search algorithm is 1.27%, and the maximum absolute error is 0.64 L/min. The experimental results show that the relative error of PID control based on the beetle antennae search algorithm is the smallest. Compared with the traditional PID control, the relative error of PID control based on the beetle antennae search algorithm is reduced by 3.12 percentage points, and the stability of the control is better than that of PID control.

**Table 3.** System flows control error.

| Speed/ (m·s⁻¹) | Theoretical Flow Rate/(L·min⁻¹) | Traditional PID | | | PID Control Based on Taurus Search Algorithm | | |
|---|---|---|---|---|---|---|---|
| | | Measured Flow Rate/ (L·min⁻¹) | Absolute Error/ (L·min⁻¹) | Relative Error/% | Measured Flow Rate/ (L·min⁻¹) | Absolute Error/ (L·min⁻¹) | Relative Error/% |
| 1.5 | 22.5 | 23.62 | 1.12 | 4.98 | 22.32 | −0.18 | −0.80 |
| 2.0 | 30.0 | 31.04 | 1.04 | 3.47 | 30.38 | 0.38 | 1.27 |
| 2.5 | 37.5 | 39.56 | 2.06 | 5.49 | 38.10 | 0.60 | 1.60 |
| 3.0 | 45.0 | 46.62 | 1.62 | 3.60 | 45.64 | 0.64 | 1.42 |

### 5.2.2. Variable Control Experiment

Variable control test collects vehicle speed by data acquisition card. The vehicle speed is measured by the angular velocity sensor, and the collected square wave signal is set and stored by the programmable signal generator. The experiment simulates the electrical signal when the vehicle speed changes through a programmable signal generator. In the experiment, the variable-flow control experiments were carried out on the traditional PID control and the PID control based on the beetle antennae search algorithm.

In the experiment, the flow rate of liquid fertilizer was measured by a flowmeter, and the experimental data were recorded after the system was stable. The experimental results of flow control under variable-speed conditions are shown in Figure 9.

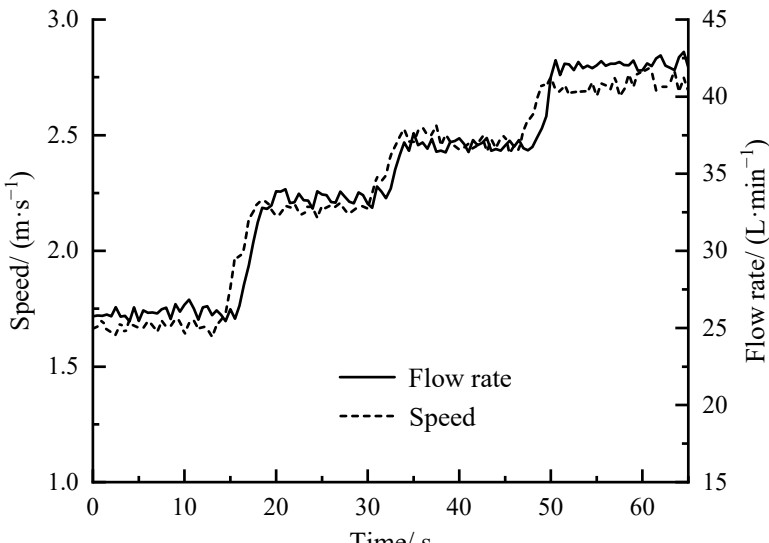

**Figure 9.** Variable control experiment.

According to Figure 8, in the variable-rate fertilization control system, the average response time of flow regulation corresponding to PID control based on beetle antennae search algorithm is 2 s. The experimental results show that the PID control based on the beetle antennae search algorithm has good response speed and flow control stability, and can meet the control requirements.

## 6. Discussion

In this study, the control system of liquid fertilizer variable-rate fertilization was studied and the control system model was established. The traditional PID control and PID based on beetle antennae search algorithm are simulated and compared, and the flow control is tested. The main conclusions are as follows:

(1)  This study uses the beetle antennae search algorithm combined with the PID control algorithm to optimize the system. The model of liquid fertilizer variable-rate fertilization control system was established. The adaptive optimization of PID parameters by using a beetle antennae search algorithm can effectively improve the performance of system transient response.

(2)  In this study, the negative feedback regulation process of the liquid fertilizer variable-rate fertilization control system was analyzed, and the control system was modeled and simulated by computer software. The results show that the response time of the traditional PID control to achieve steady-state is 2.5 s, and the overshoot is 0.014. The response time of PID control based on the beetle antennae search algorithm is 1.2 s and the overshoot is 0.006. Compared with the traditional PID control, the response time of the optimized system is reduced by 1.3 s, and the overshoot is reduced by 0.008.

(3)  The bench test results show that for the variable-rate fertilization control system in this paper, the relative error of flow rate of the traditional PID control is 4.39%, and the relative error of flow rate of the PID control based beetle antennae search algorithm is 1.27%. The actual response time of PID control based on the beetle antennae search algorithm is 2 s. Therefore, the actual control effect of PID control based on the beetle antennae search algorithm is better. The results can provide a feasible scheme for foliar fertilization or spraying control in cotton fields.

## 7. Conclusions

According to the research conclusions of this paper, it can be seen that the beetle antennae search algorithm has a good effect on the optimization of PID parameters. The research in this paper provides a feasible scheme for the study of a variable-rate fertilization

control system. The optimization steps of the beetle search algorithm used in this study are simple, and the results can be obtained by faster convergence. In the fuzzy PID control, the setting of the fuzzy control rules in the fuzzy controller needs to be carried out according to the experience of experts, which limits the use of the fuzzy controller by some people. For the third-order system in this paper, the control effect of the PID controller based on the beetle antennae search algorithm is better than that of the fuzzy PID controller. Therefore, the algorithm can meet the control requirements of complex systems, which is of great help to improve the liquid fertilizer control accuracy of the boom sprayer.

**Author Contributions:** Conceptualization, M.T. and J.L.; methodology, M.T. and J.B.; software, J.B.; validation, M.T., J.L. and J.B.; formal analysis, M.T., J.L. and J.B.; investigation, J.B.; resources, M.T.; data curation, J.B.; writing—original draft preparation, J.B.; writing—review and editing, M.T. and J.L.; visualization, J.B.; supervision, M.T. and J.L.; project administration, M.T.; funding acquisition, M.T. All authors have read and agreed to the published version of the manuscript.

**Funding:** This research was funded by the National Nature Science Foundation of China grant number 61962053 And The High-level Talent Research Project of Shihezi University grant number RCZK2018C39.

**Institutional Review Board Statement:** The study not involve humans or animals.

**Informed Consent Statement:** The study did not involve humans.

**Data Availability Statement:** The study did not report any data.

**Acknowledgments:** We are very grateful for the support provided by the College of Mechanical and Electrical Engineering and the College of Agriculture, Shihezi University.

**Conflicts of Interest:** The authors declare no conflict of interest.

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
