# Peer review of "Control System of Liquid Fertilizer Variable-Rate Fertilization Based on Beetle Antennae Search Algorithm"

_processes, doi:10.3390/pr10020357_

Round 1

Reviewer 1 Report

A control system of variable rate fertilization of nine liquid fertilizer based on beetle antennae search algorithm was proposed.
A mathematical formulation is introduced. Recent related works are investigated. The figures are of high quality.
Above some recommendations to enhance the quality of the study:
1- Why comparing the proposed algorithm with "Taurus Search Algorithm"? I suggest to choose a known optimizer (such as PSO, NSGA-III, MOEA/DD or grey wolf optimizer).
2- The use of intelligent recent methods (such as blockchain models) to resolve complex real-world problems (such as
yours) should be justified. In this regard, I suggest to refer to the following studies explaining 
the indicated statements [justification of use of intelligent methods for real-world problems]:  https://doi.org/10.1109/IWCMC.2018.8450372 and https://doi.org/10.1007/978-3-319-67910-5_2
3- Despite simulation and real experiments are proposed, few results are discussed. A more comparative analysis should be given. 
4- The conclusion does not suggests future direction/perspectives for the study.

Author Response

  1. In this paper, a PID control algorithm based on beetle antennae search algorithm is proposed. And now adds a fuzzy PID control for comparative analysis of the system simulation. (See subsection 3.3.2 in the paper)
  2. The research content of this paper can be used to optimize the actual control system. In practical problems, PID control and fuzzy PID control can be easily realized, but there are problems in the adjustment of PID parameters. Therefore, this paper proposes this PID control algorithm based on the beetle antennae search algorithm, which can solve practical control problems.
  3. The discussion and conclusion chapters have been improved. And in the conclusion chapter, the prospect of the research is presented. (See Chapter 5 and Chapter 6 in the paper)

Reviewer 2 Report

This paper proposes the application of the beetle antennae search algorithm for tunning a PID controller for a valve system of a liquid fertilizer with a variable rate. The problem is of extreme utility as is described in the introduction. The paper is mostly clear in the description of the problem and contains interesting experimental results. Therefore, I consider that the paper has significant content for publication but there are some shortcomings that the authors should address before an eventual publication.

There are two major concerns which are:

  • From my understanding, the controller is a PID and therefore it is not an adaptive control system as it is claimed in the title, and
  • The beetle antennae search algorithm, which is the core of the work, is not adequately described and there are no references for this optimization algorithm. In particular, it is not clear in figure 4 what are exactly the steps represented by the blocks "Update the next location", "random search vector" and "update the location of the beetle".

I have also the following minor concerns:

  • The language should be improved. I recommend rechecking the text with someone proficient in English or the use of grammar checking software.
  • Figure 1, although very interesting, is not completely clear, and the function of all the components is not adequately described.
  • It is not explained how the exact values for the dc motor model in equation 5 were obtained.
  • It is not clear what the parameter compensation values in equation 7 (\Delta K) represent.
  • The authors should explain what they chose the beetle antennae search algorithm over other optimization algorithms. 

Author Response

  1. Please see the attachment for details.
  2. First of all, the beetle antennae search algorithm has been described in Section 3.1. And, in the revised version, an illustration is made to Figure 4. 
  3. Added some functional descriptions to Figure 1 in Chapter 1.
  4. The revised paper gives the parameters needed to calculate the control system model. These parameters explain the source of the equations in the text. (See Table 3 in the paper)
  5. 4. The simulation analysis of fuzzy pid control is added in this paper. This will be used as the basis for selecting the PID control algorithm based on the beetle search algorithm. 

Reviewer 3 Report

The authors performed an optimization investigation for a liquid fertilizer using Beetle Antennae. I have the following comments for the authors:

  • Why the authors did not change the parameters such as set length, or the distance between the whiskers?

Author Response

The parameters of the beetle search algorithm are the optimal values that have been obtained through simulation and debugging. Therefore, the parameters after debugging are directly given in the text.

The main purpose of this paper is to compare and analyze the PID control algorithm based on the beetle search algorithm and other algorithms. Therefore, fuzzy PID control is added in the revised article for the simulation comparison of system performance. (System performance analysis of fuzzy PID control is presented in Section 3.3.2)

Round 2

Reviewer 3 Report

I accept the manuscript in its current form